# Trends in Efficacy Endpoints in Phase II Glioblastoma Trials: A Regulatory Science Analysis (FY2020–FY2022)

**DOI:** 10.3390/cancers17050855

**Published:** 2025-03-01

**Authors:** Shinya Watanabe, Makoto Maeda, Narushi Sugii, Masanobu Yamada, Yoshihiro Arakawa, Kimika Nakamura, Koichi Hashimoto, Eiichi Ishikawa

**Affiliations:** 1Department of Neurosurgery, Mito Kyodo General Hospital, Tsukuba University Hospital Mito Area Medical Education Center, Mito 310-0015, Japan; 2Institute of Medicine, University of Tsukuba, Tsukuba 305-8575, Japane-ishikawa@md.tsukuba.ac.jp (E.I.); 3Department of Pharmacy, National Cancer Center Hospital, Tokyo 104-0045, Japan; 4Tsukuba Clinical Research and Development Organization, University of Tsukuba, Tsukuba 305-8576, Japan; yamada-masanobu@md.tsukuba.ac.jp (M.Y.); koichi.hashimoto@md.tsukuba.ac.jp (K.H.); 5Comprehensive Human Sciences Research Group, Graduate School of Comprehensive Human Sciences, University of Tsukuba, Tsukuba 305-8575, Japan

**Keywords:** glioblastoma, brain tumors, efficacy endpoint, phase II, clinical trial, regulatory science

## Abstract

Efficacy evaluation based on the objective response rate (ORR) is the gold standard in other solid tumor research; in glioblastoma, ORR was used in 20% of phase II glioblastoma trials in the fiscal years (FYs) 2017–2019. We analyzed recent trends in efficacy endpoint settings for phase II glioblastoma trials conducted in FY2020–2022 and compared them with FY2017–2019 trials. Among the 101 primary endpoints from 88 trials, progression-free survival (PFS), overall survival, and PFS rates were the most common primary endpoints (PEs) at 22 (22%), 20 (20%), and 17 (17%), respectively. Time-to-event outcomes were used in 74 (73%) trials, whereas ORR was employed as a PE in only 7 trials (8%). ORR as a PE was significantly lower than in FY2017–2019 (*p* = 0.022). The diversity of efficacy endpoint settings increased in recent trials, with reduced reliance on ORR compared to earlier periods, reflecting evolving strategies to address the unique challenges of glioblastoma treatment and evaluation.

## 1. Introduction

In clinical trials before phase III, efficacy is generally evaluated using the objective response rate (ORR) [1], based on the Response Evaluation Criteria in Solid Tumors (RECIST) [2] in almost all solid tumor fields except brain tumors. This method was reported to have been used in over 90% of phase II trials to evaluate efficacy [3], serving as a surrogate marker [4]. However, due to the unique characteristics of brain tumors, such as glioblastoma, challenges remain regarding the appropriateness of RECIST for evaluating the surrogate response rate of brain tumors [3,5]. Therefore, various efficacy endpoint settings are employed in phase I and II trials for brain tumors [3,5,6]. Currently, no comprehensive reports exist to guide this issue. We focused on the status of efficacy evaluation methods that differ from those used for other solid tumors, owing to the unique characteristics of brain tumors. Glioblastoma has one of the poorest prognoses for brain tumors [7,8,9]. Therefore, the development of both treatment methods and appropriate evaluation strategies has been necessary. We focused on glioblastoma and previously studied phase II clinical trials conducted between the fiscal years (FYs) 2017 and 2019, publishing our initial findings in 2021 [3]. In this study, the most common primary efficacy endpoint (PE) settings were overall survival (OS) in 29%, ORR in 20%, progression-free survival (PFS) in 17%, and OS rate in 10% of cases [3]. However, clinical trial methodologies have likely evolved in recent years. Therefore, we studied phase II clinical trials initiated over 3 years from FY2020 to FY2022, analyzing changes and past influences on efficacy endpoint settings.

## 2. Materials and Methods

Clarivate’s Cortellis^TM^ Clinical Trial Intelligence was used to survey recent trends in efficacy evaluations in Phase II clinical trials for glioblastoma. Regarding the inclusion or exclusion criteria, a database search using the terms “glioblastoma”, “interventional study”, and “Phase II trials” for the period from 1 April 2020 to 31 March 2023 was performed. This search identified 116 Phase II trials involving glioblastomas. According to the Preferred Reporting Items for Systematic Reviews and Meta-Analyses 2020 statement [10], we planned to identify and screen trials by excluding certain studies based on specific criteria. Subsequently, the characteristics of the included trials—such as the target patient segment, types of treatment, region, organization, and other relevant factors—were summarized. Trial design characteristics were also evaluated.

The primary purpose of this study was to analyze the recent trends in efficacy endpoints and factors affecting Phase II clinical trials for glioblastoma. Therefore, we first analyzed efficacy endpoints—that is, PEs and secondary endpoints (SEs)—in the included clinical trials. Second, we analyzed the clinical trial designs for efficacy evaluation, considering not only PEs but also overall efficacy, including both Pes + SEs. Third, the trial designs for efficacy endpoints were analyzed and compared with our previous dataset between FY2017 and FY2020 [3].

To summarize baseline variables, categorical data were analyzed using frequencies and proportions, while continuous variables were described through medians, means, and standard deviations. Statistical analyses were performed using JMP version 10.0.0 (SAS: Cary, NC, USA). Comparisons of continuous variables were conducted with the Wilcoxon test, whereas categorical variables were assessed using Fisher’s exact test. A significance threshold of *p* < 0.050 was applied to all statistical evaluations.

## 3. Results

### 3.1. Baseline Characteristics

Among the 116 trials identified, 28 were excluded—19 because the enrolment registry language was not English, five due to being diagnostic trials, and four because they did not evaluate anti-tumor efficacy (Figure 1). The background information of the remaining 88 trials is summarized in Table 1. The target patient segment of the 88 trials consisted of half newly diagnosed patients and half recurrent patients. The largest proportion of these trials tested pharmaceutical products, followed by combination therapies, biological products, medical devices, radiotherapy, supplements, and treatment procedures. Most of the studies were conducted in the United States, followed by China. Most research was conducted in academic settings, followed by collaborations between academia and companies, companies alone, between academia and governments, and governments alone. The characteristics of the trial design are presented in Table 2. The median number of trial arms was one, with a maximum of four. The majority of the trials had one study site, with a maximum of 545 sites. The median planned trial duration was 38 months, ranging from 8 to 118 months. The median enrolment of subjects in each trial was 39 patients, ranging from 1 to 640 patients.

### 3.2. Primary and Secondary Efficacy Endpoints

In all 88 trials, a total of 101 efficacy PEs were identified. The median number of efficacy PEs was 1 (range: 0–7): 12 trials had none, 60 trials had 1, 12 trials had 2, 2 trials had 3, 1 trial had 4, and 1 trial had 7. Categories for all efficacy PEs are summarized in Table 3. PFS, OS, and PFS rates were approximately 20%; moreover, the immunological marker/tumor cell rate was 13%, the OS rate was 11%, and the ORR was 7%. Estimation methods for ORR included the Response Assessment in Neuro-Oncology (RANO) [11] and the Immunotherapy Response Assessment in Neuro-Oncology (iRANO) [12].

Conversely, a total of 209 SEs were found to be efficacious. The median number of efficacy SEs was three (range: 1–12): 2 trials had 12 SEs, 1 trial had 11, 1 trial had 8, 8 trials had 7, 6 trials had 6, 6 trials had 5, 14 trials had 4, 15 trials had 3, 13 trials had 2, and 12 trials had 1. The items for all efficacy SEs are summarized in Table 4. OS was the most common outcome (approximately 15%), followed by PFS, quality of life, and ORR. The estimated ORR was 15 for RANO, 8 for iRANO, 2 for RECIST, 1 for other methods, and 6 were not available.

Immunological markers and tumor cell parameters were assessed as part of the exploratory endpoints. Specifically, immunological markers included T-cell responses, tumor-infiltrating T cells, levels of T regulatory cells, and blood data such as the blood glucose-to-ketone ratio. Tumor cell-related assessments included tumor cell death and tumor cell proliferation measurements. These parameters were analyzed to evaluate immune system activation and tumor biology in response to treatment. Details of neurological outcomes were assessed using the Neurologic Assessment in Neuro-Oncology scale [13] and clinical activity.

### 3.3. Clinical Trial Settings for Efficacy Endpoints

Regarding the clinical trial design for efficacy endpoints (Table 5), multiple efficacy endpoints were included in 32% of the trials; however, ORR was set as an efficacy endpoint in only 8% of the trials. Conversely, the time-to-event (TTE) outcomes were included in three-quarters of the trials, with the control arm present in 18% of the trials. Regarding efficacy assessment, including both efficacy PEs and SEs (Table 6), multiple endpoints were included in 99% of the trials, ORR was set in approximately half of the trials, TTE outcomes were set in 92% of the trials, and control arms were present in only 22% of the trials.

### 3.4. Comparison of Efficacy Primary Endpoint with Clinical Trials Started in FY2017–2019

In phase II clinical trials for glioblastoma conducted between FY2017 and FY2019, the most common PE settings were OS at 29%, ORR at 20%, PFS at 17%, and OS rate at 10% [3]. Compared to clinical trials initiated in FY2017–2019 (Table 7), those conducted in FY2020–2022 demonstrated a statistically significant reduction in ORR usage for PE (Figure 2: *p* = 0.022).

## 4. Discussion

In the analysis of trials in FY2020–2022, the PE setting was highly variable—PFS, OS, and PFS rates were approximately 20%—moreover, the immunological marker/tumor cells, OS rate, and ORR were 13%, 11%, and 7%, respectively. Furthermore, a greater variety of efficacy endpoints were observed in SEs. Clinical trial designs for the efficacy of PEs and PEs + SEs included multiple efficacy endpoints in 32% and 99% of the trials, respectively, with ORR being set as an efficacy endpoint in only 8% and 45% of the trials, respectively. When examining the overall efficacy endpoints of PE and SEs, many trials had multiple efficacy endpoints. TTE outcomes were included in 72% and 92% of the clinical trials, while control arms were present in only 18% and 22% of the trials, respectively. PE types have become increasingly diverse in recent years, with a significant decrease in the proportion of ORRs defined as PEs. In comparison with clinical trials conducted between FY2017 and FY2019 and FY2020 and FY2022, those in FY2020–2022 were significantly less likely to set ORR as a PE (*p* = 0.022).

### 4.1. Were Recent Endpoint Settings Affected by the Study Design of the Trial?

This study highlights the unique features of efficacy evaluation in glioblastoma in recent years, such as the diverse efficacy endpoints and low ORR setting rates that differ from those in other solid tumor fields. Previously, we highlighted the uniqueness of glioblastoma clinical trials: target lesions after standard surgical resection are generally irregular due to anatomic limitations [3], image modifiers such as pseudoprogression [5], or radiation necrosis [14], following standard multidisciplinary treatment. Additional hurdles include diverse tumor classifications [15] and small sample sizes due to the rarity of these conditions [16,17]. Our research into clinical trials conducted in FY2020–2022 revealed the use of unique efficacy evaluation methods significantly different from those in other solid tumor fields. Therefore, we tried to identify potential factors relevant to the recent efficacy endpoints setting, that is, diverse efficacy endpoints and low ORR, in clinical trials (Table 6).

#### 4.1.1. Type of Clinical Trial Item

The settings of phase II clinical trials of products previously approved for glioblastoma included a high number of pharmaceutical drugs, numerous studies targeting recurrent glioblastoma, predominantly open-label and uncontrolled studies, with PFS or OS rates frequently used as endpoints [18,19,20,21,22]. Regarding the type of clinical trial item, we examined whether the trial item in our study was pharmaceutical only, which led to observable differences (Table 8). The number of efficacy PEs was slightly higher in drug trials (*p* = 0.053); however, the number of SEs did not differ between the two groups (*p* = 0.13). The rates of ORR set as PE were low, at 8% for both groups (*p* = 1.00), and remained under 50% when including SEs (*p* = 1.00). The establishment of efficacy endpoints for TTE showed high rates for both PEs at 70% (*p* = 0.48) and for SEs at >90% (*p* = 0.48). In general, a few differences were observed in efficacy evaluation settings depending on the investigational product.

#### 4.1.2. Target Patient Segment

Previous regulatory science studies have shown ORR setting as a PE in phase II clinical trials targeting glioblastoma in 4% of the trials for newly diagnosed glioblastoma and 39% for recurrent glioblastoma, indicating a tendency for ORR to be more prevalent in clinical trials targeting recurrent glioblastoma [3]. In this study, regarding the factor of the target patient segment (newly diagnosed vs. recurrent; Table 9), the number of PEs was significantly higher in recurrent glioblastoma (*p* = 0.046). ORR settings as a PE were higher for recurrent glioblastoma (0% vs. 17%; *p* = 0.050), with a significantly higher overall efficacy assessment (PE + SE) (27% vs. 64%; *p* = 0.0014). Regarding the establishment of efficacy endpoints for TTE, the rates were no different between the PE group (*p* = 0.33) and the SE group (*p* = 0.21).

There was a statistically significant difference in the number of PEs for recurrent glioblastoma, with ORR more frequently used. Various factors may explain why ORR is set as both PE and SE in recurrent cases compared to newly diagnosed cases. The hypotheses included pseudoprogression [5] and radiation necrosis [14]. These conditions, which may arise following initial standard treatment, can complicate the determination of recurrence through imaging evaluations, as defined by the inclusion criteria. In addition, initial standard treatment involves surgical removal; imaging evaluation may be difficult, although surgery is often not performed after recurrence. Therefore, the primary reason for this difference is that ORR setting may be adapted more frequently in cases of recurrent disease than in newly diagnosed cases. Additionally, when analyzing the breakdown of TTE in PE across clinical trials targeting new/recurrent cases, the distributions are as follows: PE was reported at 27% and 24% for PFS, 30% and 17% for OS, 11% and 29% for PFS rate, and 9% and 14% for OS rate, for new and recurrent cases, respectively. In addition, 6 (13%) of 44 trials for new cases and 2 (5%) of 42 recurrent cases were randomized double-blinded clinical trials. The control group in all but one of the trials for new cases was the Stupp regimen [23] + placebo group, and the control group in the recurrent cases was the placebo group. Further investigations are required to clarify these findings.

#### 4.1.3. Trial Design: Randomized Double-Blinded Clinical Trial or Not

In past phase II trials of approved drugs, open-label single-arm trials were commonly used [19,20,21,22]. No regulatory science studies have examined differences in endpoint settings due to differences in clinical trial designs. Regarding trial design (randomized double-blind clinical trial; Table 10), the number of efficacy PEs was slightly higher in randomized double-blind trials (*p* = 0.10), but the number of SEs was significantly lower (*p* = 0.016). Regarding ORR, there was no difference in its frequency as PE (*p* = 1.00); however, the overall evaluation, including SEs, tended to be higher in non-double-blind studies (*p* = 0.070). There was no difference in TTE endpoints, either for PE (*p* = 1.00) or the overall evaluation, including SEs (*p* = 0.49). The number of SEs was significantly lower in randomized double-blind clinical trials (*p* = 0.016). The frequency of ORR was higher in non-double-blind studies across all efficacy evaluations, including SEs (*p* = 0.070).

### 4.2. What Are the Reasons for the Changes in Trial Endpoints, and the Pros and Cons of Those Changes?

In our study, the frequency of using ORR as a PE was significantly reduced compared to that of the previous study, while immunological markers/tumor cells were relatively high (13%) and the PFS and PFS rates increased. The landscape of anticancer drug development has undergone a significant transformation in recent years, primarily driven by advancements in molecularly targeted therapies. Notably, the emergence of immune checkpoint inhibitors and treatments designed to inhibit key driver mutations has revolutionized therapeutic strategies, reshaping the approach to cancer treatment. Anticancer guidelines state that although RECIST-based response assessment criteria remain the standard for solid tumors, appropriate criteria should be used for each investigational drug following scientific advances [1]. The reasons for the further decline in the proportion of ORRs set as PE and the change in diversification of settings revealed in our study are possibly due to the changes in the developmental environment described earlier, which have contributed to this shift. One meta-analysis identified a strong correlation between treatment effects on ORR and PFS in randomized clinical trials investigating agents targeting oncogene-driven cancers, a weaker correlation was observed between ORR and OS [24]. Batichi et al. showed OS reproducibility of dendritic cell vaccine trials targeting cytomegalovirus in glioblastoma [25]. The pros and cons of increasing the diversification of effectiveness evaluations are as follows: One of the pros is that it can address the issue that it is difficult to properly evaluate treatments that involve temporary tumor growth (pseudoprogression), such as immune checkpoint inhibitors, using ORR alone. Conversely, a significant drawback in the field of glioblastoma, where historical data are insufficient compared to other solid tumors and the TTE data are lacking, is the reorganization of disease categories. Specifically, the recent revisions in the World Health Organization (WHO) classification [15] will likely complicate the efficacy evaluation in the future. This complexity may hinder the exploratory role of phase II trials.

What are the future trends in evaluation methods? One possible trend is the adoption of the new RANO criteria, published as a revised version called RANO 2.0 [26], for the clinical evaluation of gliomas. RANO 2.0 criteria represent a significant revision of the original RANO framework, addressing critical limitations in glioblastoma response assessment. The updated criteria refined imaging evaluation by introducing stricter definitions, clearer detailed guidelines for enhancing or non-enhancing tumor, and keeping cross-sectional diameter use and option use of volume, despite growing interest in 3D volumetric measurements [26], which might enhance reproducibility in clinical trials [27]. A key modification is the introduction of new response categories, including “Minor Response”, which help differentiate ambiguous cases and reduce premature progression classification [26]. Another critical update is the formal integration of iRANO into RANO 2.0, recognizing the growing role of immune checkpoint inhibitors and other immunotherapies in glioblastoma. This change ensures that atypical response patterns, such as delayed tumor shrinkage or pseudoprogression, are adequately accounted for [28,29]. The adoption of RANO 2.0 is expected to significantly impact endpoint selection in phase II glioblastoma trials. Future studies should assess the real-world impact of RANO 2.0 on clinical trial outcomes and regulatory decision-making, particularly in light of its implications for immunotherapy evaluation. In our current study, the frequency of ORR was reduced, but it will be necessary to observe whether the frequency of OS and PFS will increase in future phase II clinical studies. In addition, challenges have emerged in interpreting the differences between the old and new versions of the RANO evaluation methods. Changes in the clinical evaluation of brain tumors are expected to continue, and further research will be conducted to address these developments.

### 4.3. Strengths and Limitations of the Study

This study uniquely analyzed recent trends in efficacy endpoint and factors affecting phase II clinical trials of glioblastoma. However, this study had some limitations. First, it relied on a single database, as trials in languages other than English were excluded. Second, we could not compare our results with those of other studies because there was little supporting literature available. Therefore, internal evidence is being continuously accumulated. This study focused on the evolution of phase II trial design and did not provide detailed tracking of the transition to phase III trials. However, this is an important issue to consider in future studies. Thus, we plan to investigate the rate of progression to phase III in future studies. Third, due to the significant heterogeneity, we did not consider how variations in the clinical trial protocols (e.g., combination therapies, treatment regimens) might have influenced the choice of efficacy endpoints. Such analysis could provide further insights into the rationale behind selecting certain endpoints, which we plan to conduct in our future study. Additionally, new WHO brain tumor guidelines [15,30] have been issued; therefore, evidence based on this classification should be accumulated for future analyses. Glioblastomas are unique in many ways; biological factors, such as the blood–brain barrier and the unique tumor and immune microenvironment, represent significant challenges in the development of novel therapies. Innovative clinical trial designs incorporating biomarker-enrichment strategies are needed to improve the outcomes of patients with glioblastoma [9]. Future studies may need to explore methods for individualized efficacy evaluation based on new classifications, such as gene mutations.

## 5. Conclusions

Recently, the PE settings have been highly variable; furthermore, SE settings exhibited a greater variety of efficacy endpoints with multiple efficacy endpoints. Surprisingly, ORR settings such as PE and SE were shown in only 8% and 45% of recent trials, respectively. PE types have become increasingly diverse in recent years. In comparison with clinical trials between FY2017 and FY2019, those in FY2020–2022 were revealed to have a statistically lower frequency of ORR being set as a PE. The results of this study may influence more appropriate efficacy evaluation in future phase II clinical trials targeting glioblastoma. In addition, the new WHO classification was revised, and the new RANO classification was recently proposed; thus, we believe that data on ORR should at least be included as an SE, but continued discussion is essential.

## Figures and Tables

**Figure 1 cancers-17-00855-f001:**
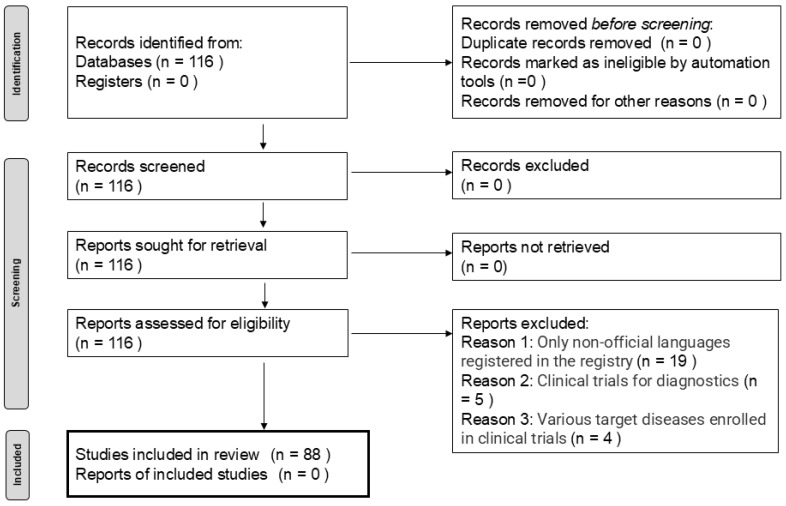
Identification of studies via databases and registers for phase II clinical trials targeting glioblastoma.

**Figure 2 cancers-17-00855-f002:**
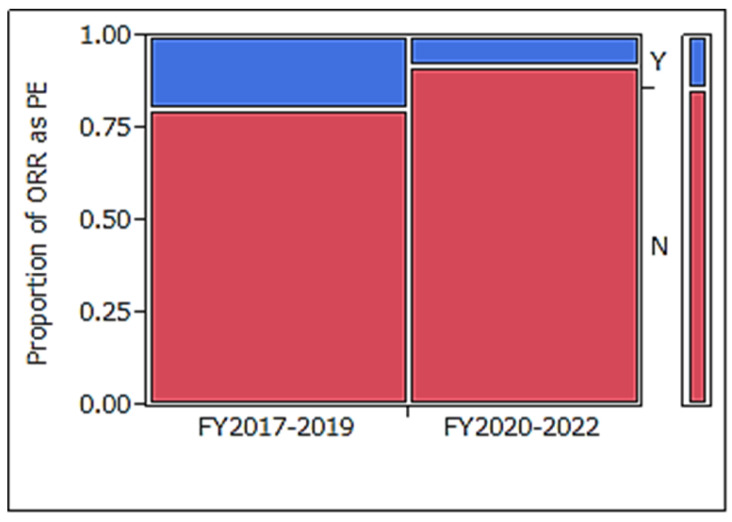
Comparison of ORR proportion as PE: FY2017–2019 vs. FY2020–2022. The percentage of trials in which ORR was included as PE was 20% in FY2017–2019, decreasing to 7% in FY2020–2022. ORR: objective response rate, PE: primary endpoint, FY: fiscal year, Y: yes, N: no.

**Table 1 cancers-17-00855-t001:** Baseline characteristics on the phase II clinical trials included in this study (N = 88).

Item	Category	Number of Trials (%)
Patient segmentation	Newly diagnosed	44 (50)
	Recurrent	42 (48)
	Both	2 (2)
Item category	Pharmaceutical	51 (58)
	Multiple combinations	11 (13)
	Biological product	8 (9)
	Medical device	6 (7)
	Radiotherapy	5 (6)
	Supplement	3 (3)
	Treatment procedure change	2 (2)
	Others	2 (2)
Region *	USA	29 (33)
	China	21 (24)
	Germany	4 (5)
	Italy	3 (3)
	Japan	2 (2)
	Canada	2 (2)
	France	2 (2)
	Israel	2 (2)
	Switzerland	2 (2)
	Others **	18 (20)
Organization(s)	Academia	48 (55)
	Academia + company	21 (24)
	Company	12 (14)
	Academia + government	4 (5)
	Government	3 (3)

* Not available in 3 trials, ** Each single country.

**Table 2 cancers-17-00855-t002:** Baseline characteristics of trial design on phase II clinical trials included in this study (N = 88).

Category	Numbers (Median, Min–Max)
Trial arm numbers (arm)	1, 1–4
Trial sites (sites)	1, 1–545
Planned trial duration * (months)	38, 8–118
Enrolment patients (persons)	39, 1–640

Min, minimum; Max, maximum. * This is the maximum expected period, as the trial may actually end early.

**Table 3 cancers-17-00855-t003:** Categories of primary endpoints (N = 101).

Category	Numbers of Endpoint (%)
PFS	22 (22)
OS	20 (20)
PFS rate	17 (17)
Immunological marker/tumor cell	13 (13)
OS rate	11 (11)
ORR	7 * (7)
DOR	3 (3)
Neurological outcome	3 (3)
Anxiety	2 (2)
RFS	1 (1)
DRR	1 (1)
Others	1 (1)

PFS: progression-free survival; OS: overall survival; ORR: objective response rate; DOR: duration of response; RFS: relapse-free survival; DRR: durable response rate; RANO: Response Assessment in Neuro-Oncology; iRANO: Immunotherapy Response Assessment in Neuro-Oncology. * The ORR assessment methods for all 7 were 6 RANO and 1 iRANO.

**Table 4 cancers-17-00855-t004:** Categories of secondary endpoints * (N = 299).

Category	Numbers of Endpoint (%)
OS	45 (15)
PFS	44 (15)
QOL	43 (14)
Immunological marker/tumor cell	35 (12)
ORR	32 ** (11)
Neurological outcome	15 (5)
PFS rate	13 (4)
Cognitive function	12 (4)
DCR	9 (3)
OS rate	8 (3)
DOR	7 (2)
Volume	5 (2)
Anxiety	2 (1)
RFS	2 (1)
EFS	2 (1)
Others	25 (8)

OS: overall survival; PFS: progression-free survival; QOL: quality of life; ORR: objective response rate; RANO: Response Assessment in Neuro-Oncology; iRANO: Immunotherapy Response Assessment in Neuro-Oncology; RECIST: Response Evaluation Criteria in Solid Tumors; DCR: disease control rate; DOR: duration of response; RFS: relapse-free survival; EFS: event-free survival. * Analysis of 78 trials, excluding 10 trials in which SE was not disclosed. ** The ORR assessment methods for all 32 cases included 15 RANO, 8 iRANO, 2 RECIST, 1 “Other”, and 6 “Not available”.

**Table 5 cancers-17-00855-t005:** Summary of clinical trials design aspects on primary endpoints only (N = 88).

Design Type of Trials	Number of Trials (%)
Multiple efficacy endpoints	28 (32)
ORR *	7 (8)
TTE outcome	63 (72) **

ORR, overall response rate; TTE, time-to-event; RANO; Response Assessment in Neuro-Oncology; iRANO, Immunotherapy Response Assessment in Neuro-Oncology. * The ORR assessment method for all 7 trials was RANO in 6 trials and iRANO in 1 trial. ** Among the 63 trials, 16 trials had a control arm.

**Table 6 cancers-17-00855-t006:** Summary of clinical trial design aspects on primary endpoints + secondary endpoints (N = 78 *).

Design Type of Trials	Number of Trials (%)
Multiple efficacy endpoints	77 (99)
ORR **	35 *** (45)
TTE outcome	72 (92) ****

ORR: overall response rate; TTE: time-to-event; RANO: Response Assessment in Neuro-Oncology; iRANO: Immunotherapy Response Assessment in Neuro-Oncology; RECIST: Response Evaluation Criteria in Solid Tumors. * Analysis of 78 trials, excluding 10 trials where the secondary endpoint was not disclosed. ** There were duplicates in trials with multiple ORRs. *** The ORR assessment methods for all 35 trials were RANO in 19 trials, iRANO in 9 trials, RECIST in 2 trials, “Other” in 1 trial, and “Not available” in 6 trials. **** Among the 72 trials, 17 had a control arm.

**Table 7 cancers-17-00855-t007:** Comparison of efficacy PE between clinical trials started in FY2017–2019 and FY2020–2022.

	FY2017–2019	FY2020–2022	*p*-Value
Number			0.13 **
median (Min–Max)	1, 0–5	1, 0–7	
average ± SD	1.21 ± 0.57	1.15 ± 0.93	
Types of endpoints	OS 29%	PFS 22%	0.022 ****
	ORR 20%	OS 20%	
	PFS 17%	PFS rate 17%	
	OS rate 10%	Immunological/tumor marker 13%	
	PFS rate 9%	OS rate 11%	
	Others 14%	ORR 7%	
		DOR 3%	
		Neurological outcome 3%	
		Anxiety 2%	
		RFS 1%	
		DRR 1%	
		Others 1%	
Proportion of RANO + iRANO in ORR *	13 **/16 (81%)	7 ***/7 (100%)	0.52 *****

FY: fiscal year; PE: primary endpoint; min: minimum; max: maximum; SD: standard deviation; RANO: Response Assessment in Neuro-Oncology; iRANO: Immunotherapy Response Assessment in Neuro-Oncology; ORR: overall response rate; PFS: progression-free survival; OS: overall survival; DOR: duration of response; DCR: disease control rate; QOL: quality of life. * Among 20 clinical trials conducted in FY2017–2019, 4 clinical trials without details of the ORR method were excluded. ** RANO: 11, iRANO: 2, RECIST: 3. *** RANO: 6, iRANO: 1, **** Fisher’s two-tailed test: ORR (Yes or No) ***** Fisher’s two-tailed test: RANO and iRANO (Yes or No).

**Table 8 cancers-17-00855-t008:** Differences in efficacy endpoint settings: whether the item categories are pharmaceuticals alone or not.

		Pharmaceuticals Alone	Not	*p*-Value
Number median (Min–Max) average ± SD	efficacy PE	1 (0–7) 1.18 ± 1.03	1 (0–4) 1.11 ± 0.77	0.053
	efficacy SE *	3 (1–11) 3.50 ± 2.15	4 (1–12) 4.41 ± 2.78	0.13
ORR	efficacy PE	8% (4/51)	8% (3/37)	1.00
	efficacy PE + SE *	45% (20/44)	44% (15/34)	1.00
TTE outcome	efficacy PE	75% (38/51)	68% (25/37)	0.48
	efficacy PE + SE *	91% (40/44)	94% (32/34)	0.48

PE: primary endpoint; Min: minimum; Max: maximum; SD: standard deviation; SE: secondary endpoint; ORR: overall response rate; TTE: time-to-event. * Analysis of 78 studies, excluding 10 studies for which SE did not disclose information.

**Table 9 cancers-17-00855-t009:** Differences in efficacy endpoint settings: whether the target patient segment is new or recurrent *.

		Newly Diagnosed	Recurrent	*p*-Value
Number median (Min–Max) average ± SD	efficacy PE	1 (0–4) 0. 98 ± 0.73	1 (0–7) 1.33 ± 1.07	0.046
	efficacy SE **	4 (1–11) 3.90 ± 2.17	3 (1–12) 3.81 ± 2.79	0.42
ORR	efficacy PE	0	17% (7/42)	0.050
	efficacy PE + SE **	27% (11/41)	64% (23/36)	0.0014
TTE outcome	efficacy PE	68% (30/44)	79% (33/42)	0.33
	efficacy PE + SE **	88% (36/41)	97% (35/36)	0.21

PE: primary endpoint; Min: minimum; Max: maximum; SD: standard deviation; SE: secondary endpoint; ORR: overall response rate; TTE: time-to-event. * A study of 86 trials, excluding 2 trials that included both first-episode and relapse, ** Analysis of 77 studies, excluding 9 studies for which SE information was not disclosed.

**Table 10 cancers-17-00855-t010:** Differences in efficacy endpoint settings: whether the trial design is a randomized double-blinded clinical trial or not.

		Randomized Double-Blinded Clinical Trial	Not	*p*-Value
Number median (Min–Max) average ± SD	efficacy PE	1 (1–2) 1.38 ± 0.52	1 (0–7) 1.13 ± 0.96	0.10
	efficacy SE *	2 (1–4) 2.13 ± 1.13	4 (1–12) 4.10 ± 2.50	0.016
ORR	efficacy PE	0	9% (7/80)	1.00
	efficacy PE + SE *	13% (1/8)	49% (34/70)	0.070
TTE outcome	efficacy PE	75% (6/8)	71% (57/80)	1.00
	efficacy PE + SE *	88% (7/8)	93% (65/70)	0.49

PE: primary endpoint; Min: minimum; Max: maximum; SD: standard deviation; SE: secondary endpoint; ORR: overall response rate; TTE: time-to-event. * Analysis of 78 studies, excluding 10 studies for which SE information was not disclosed.

## Data Availability

The original contributions presented in the study are included in the article, and further inquiries can be directed to the corresponding author.

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
