# Peer review of "Trends in Efficacy Endpoints in Phase II Glioblastoma Trials: A Regulatory Science Analysis (FY2020–FY2022)"

_cancers, 2025, doi:10.3390/cancers17050855_

Round 1
Reviewer 1 Report
Comments and Suggestions for Authors
The Authors analyzed trends in efficacy endpoints for phase 2 trials for glioblastoma. The subject of the article is interesting for the readers that want to design future studies or just correctly interpret recent studies. The analysis is adequate in the methods and presentation of results. I think that the interpretation of results could be improved. Even if the Authors suggest some interpretation of data, the readers would like to understand what would be the best efficacy endpoint for a phase II trial. Moreover, RANO 2.0 criteria should be furtherly discussed.
Author Response
Reviewer1
The Authors analyzed trends in efficacy endpoints for phase 2 trials for glioblastoma. The subject of the article is interesting for the readers that want to design future studies or just correctly interpret recent studies. The analysis is adequate in the methods and presentation of results. I think that the interpretation of results could be improved. Even if the Authors suggest some interpretation of data, the readers would like to understand what would be thebest efficacy endpoint for a phase II trial. Moreover, RANO 2.0 criteria should be furtherly discussed.
Our reply: We deeply appreciate your positive evaluation of our study. We amended the discussion regarding the interpretation of the results to include an expanded discussion of the best efficacy endpoints going forward, including the impact of RANO 2.0.
Reviewer 2 Report
Comments and Suggestions for Authors 1)It would be convenient how many studies there are of first and second line pharmaceuticals and the proportion of them with end point main overall survival.2)It would be convenient to clarify how many studies in recurrent or first-line disease have been randomized and which control arm was used in recurrent disease.
3)It would be convenient to know how many studies have moved on to phase III studies.
Author Response
Reviewer2
1)It would be convenient how many studies there are of first and second line pharmaceuticals and the proportion of them with end point main overall survival.
2)It would be convenient to clarify how many studies in recurrent or first-line disease have been randomized and which control arm was used in recurrent disease.
Our reply: We deeply appreciate your helpful suggestions. Thank you for your comments regarding the important points to consider when considering the differences in study design between newly diagnosed and relapsed cases. We have added data and provided a discussion regarding the two points you commented on in “4.1.2. Target patient segment section.”
3)It would be convenient to know how many studies have moved on to phase III studies.
Our reply: We deeply appreciate your helpful comments. This study focused on the evolution of Phase II trial design and did not provide detailed tracking of the transition to Phase III trials. In addition, because this is a Phase II clinical trial that began recently, not enough cases to determine whether or not the study has progressed to Phase III are available. However, this is an important issue to consider in future studies. We added the following: “We plan to investigate the rate of progression to Phase III in future studies” to the Limitations section 
Reviewer 3 Report
Comments and Suggestions for Authors
The authors describe how endpoints in trials of new treatments for glioblastoma have changed in the last decade, with objective response rate less frequently used as a primary endpoint. As presented, it isn't always clear if the data describe the *how often* ORR is used as a PE, or what the OR rates actually *were* in studies. This needs clarification.
In addition, the authors should suggest reasons for the changes in trial endpoints, and the pros and cons of those changes.
Author Response
Reviewer3
The authors describe how endpoints in trials of new treatments for glioblastoma have changed in the last decade, with objective response rate less frequently used as a primary endpoint. As presented, it isn't always clear if the data describe the *how often* ORR is used as a PE, or what the OR rates actually *were* in studies. This needs clarification.
In addition, the authors should suggest reasons for the changes in trial endpoints, and the pros and cons of those changes.
Our reply: We deeply appreciate your helpful suggestions. Based on the comments we received, we have reconsidered and more clearly stated the specific numerical values of the ORR and OS rates in previous studies in the Summary, Abstract, and main text. We have also reorganized the "Results" section of the main text to make the specific numerical values easier to understand. We hope that this revision will further strengthen the relevance of this paper.
In addition, in section 4.2, we have described the reasons for recent changes in efficacy endpoints and their advantages and disadvantages.
Reviewer 4 Report
Comments and Suggestions for Authors
This is a potentially interesting paper about the status quo of phase II glioblastoma trials between fiscal years (FY) 2020 and 2022 worldwide, using Clarivate’s Cortellis Clinical Trial Intelligence database, which was then compared with the status quo ante between FY 2017-2019. This paper has many hints about where GBM clinical trials came from and where they are likely to go in the future. I recognize many favorable points about the paper, but I cannot help expressing the following concerns about it.
#’The median trial duration was 38 months, ranging from 8 to 118 months’ (line 102-103) or Table 2:
According to the authors, the clinical trials targeted in the present study are those initiated between FY 2020 and 2022, right? The median trial duration, 38 months (=3 years or so), may be okay, but then how about the ‘118 months’??? ‘118 months’ equals almost 10 years, doesn’t it? It is now 2025, the trials initiated between FY 2020 and 2022, and yet the median trial duration was almost 10 years? How could it be so? I could not understand this point. A sufficient explanation is necessary for the readers who may have the same question as this reviewer.
#’hematological marker/tumor cell’ or ‘Neurological outcome’ or ‘Cognitive function’ (text or Table 3): For example, what exactly does ’hematological marker/tumor cell’ mean? The normalization of a hematological marker or the disappearance of tumor cells? A more detailed explanation should be given, at least in the text.
#Figure 2
It says ‘decreasing to 7% in FY2020-2022. I believe, according to the text or Table 4, this ‘7%’ should be ‘8%’.
#line 190-202: First the text says about Table 5, and then it goes back to Table 4; that is, the text goes forwards and then backwards, being mixed up with each other. It is very difficult to follow the descriptions. Not only in here but potentially somewhere else in the text, the descriptions should follow up the order of appearance of Tables or Figures; otherwise, the readers cannot follow what the authors really want to say.
#line 203 and thereafter: It says quite abruptly, ‘4.1. Were endpoint settings affected by the study design of the trial?’ and a series of questions follows (4.1.1. and so). Does this happen to be the result or the discussion? It seems that what should be in the result and what should be in the discussion are being mixed up. I admit that what the authors address here is potentially interesting and important, but because of the above, it is very difficult, after all, to catch up what the authors really want say in this paper. What the authors want to say/show using this study should be ordered in the right (at least better) way so that its content can be more easily and nicely conveyed to the readers.
#Generally, this Ms is well-written with a good command of English. But a few potentially unfavorable points or editing errors, although they may be minor, are noted. Please check the Ms again.
Ex:
-line 96 ‘bioproducts’ whereas Table 1 ‘Bio products’: Whether a space is present or not? A term meaning the same thing should be identical.
-line 103 ‘enrollment’ whereas Table 1 ‘Enrolment’: I know both ‘enrollment’ and ‘Enrolment’ are grammatically correct. But just one form of spelling for a thing is more advisable???
-line 122 to 123: (RANO) [11] in 6 and (iRANO) [12] in 1 (?) (*This is judging from Table 3-a)
-Table 3-a, b:
Ex in Table 3-a: *The ORR assessment methods for all seven were RANO in six and iRANO in one. (?)
-Neurooncology or Neuro-oncology (?) throughout the Ms including the tables.
-line 193: Pes >>> PEs
-etc., throughout the Ms
Author Response
Reviewer4
This is a potentially interesting paper about the status quo of phase II glioblastoma trials between fiscal years (FY) 2020 and 2022 worldwide, using Clarivate’s Cortellis Clinical Trial Intelligence database, which was then compared with the status quo ante between FY 2017-2019. This paper has many hints about where GBM clinical trials came from and where they are likely to go in the future. I recognize many favorable points about the paper, but I cannot help expressing the following concerns about it.
Our reply: We deeply appreciate your positive evaluation of our study and your helpful suggestions. We have revised the manuscript in a point-by-point manner based on your suggestions.
#’The median trial duration was 38 months, ranging from 8 to 118 months’ (line 102-103) or Table 2:
According to the authors, the clinical trials targeted in the present study are those initiated between FY 2020 and 2022, right? The median trial duration, 38 months (=3 years or so), may be okay, but then how about the ‘118 months’??? ‘118 months’ equals almost 10 years, doesn’t it? It is now 2025, the trials initiated between FY 2020 and 2022, and yet the median trial duration was almost 10 years? How could it be so? I could not understand this point. A sufficient explanation is necessary for the readers who may have the same question as this reviewer.
Our reply: We deeply appreciate your helpful comment, which helped us make the argument of the paper easier to understand. The duration of this portion of the clinical trial is simply the data on the planned clinical trial period registered in the registry. Therefore, the description in this section (main text and Table 2) has been described as the "planned" trial duration. In addition, we have added a footnote in Table 2 to indicate that it is the maximum expected period, as the trial may actually end early.
#’hematological marker/tumor cell’ or ‘Neurological outcome’ or ‘Cognitive function’ (text or Table 3): For example, what exactly does ’hematological marker/tumor cell’ mean? The normalization of a hematological marker or the disappearance of tumor cells? A more detailed explanation should be given, at least in the text.
Our reply: We deeply appreciate your insightful comment. After carefully considering your comments, we have added more detailed information about blood markers/tumor cells, neurological outcomes, and cognitive function in the main text in “3.2 Efficacy primary & secondary endpoints.”
Thank you for your valuable comment. We agree that the term “hematological marker/tumor cell” was too broad and have now clarified the specific markers included in this category. In the revised manuscript, we specify that the hematological markers assessed were immunogenicity, T cell responses, tumor-infiltrating T cells, levels of T regulatory cells (Tregs), and blood data (blood glucose-to-ketone ratio). Tumor cell-related assessments included tumor cell death and tumor cell proliferation. This clarification has been incorporated into the Methods and Results sections of the manuscript.
We acknowledge that “hematological marker” may not be the most precise term to describe the immune-related biomarkers analyzed in our study. To improve clarity, we have revised the terminology to “immunological markers,” which more accurately reflects the evaluated parameters, including immunogenicity, T cell responses, tumor-infiltrating T cells, and T regulatory cells (Tregs). This revision ensures consistency with standard classifications in immuno-oncology research.
We hope that this revision will make this paper more valuable.
#Figure 2
It says ‘decreasing to 7% in FY2020-2022. I believe, according to the text or Table 4, this ‘7%’ should be ‘8%’.
Our reply: We deeply appreciate your helpful comment. To summarize, with regards to the 101 primary efficacy endpoints set in the 88 clinical trials, when examining the simple efficacy endpoint proportion, the total number of 101 was used as the denominator, and there were seven ORRs, which comes to 7/10, or 7%. We also considered the study design, in which case the denominator would be the number of clinical trials, which is 88, and the number of clinical trials for which the ORR was designed would be the numerator, which would be 7, giving a value of 8%.
In order to clarify the above situation and make the argument easier to understand, we have revised the results section by organizing it into chapters such as “3.2 Efficacy primary and secondary endpoints,” “3.3 Clinical trial settings for efficacy endpoints,” and “3.4 Clinical trial settings for efficacy endpoints.” We hope that this revision will further strengthen the paper.
#line 190-202: First the text says about Table 5, and then it goes back to Table 4; that is, the text goes forwards and then backwards, being mixed up with each other. It is very difficult to follow the descriptions. Not only in here but potentially somewhere else in the text, the descriptions should follow up the order of appearance of Tables or Figures; otherwise, the readers cannot follow what the authors really want to say.
Our reply: We deeply appreciate your helpful suggestions. According to your suggestions, we have revised the tables and figures to be presented in the order in which they appear in the text.
#line 203 and thereafter: It says quite abruptly, ‘4.1. Were endpoint settings affected by the study design of the trial?’ and a series of questions follows (4.1.1. and so). Does this happen to be the result or the discussion? It seems that what should be in the result and what should be in the discussion are being mixed up. I admit that what the authors address here is potentially interesting and important, but because of the above, it is very difficult, after all, to catch up what the authors really want say in this paper. What the authors want to say/show using this study should be ordered in the right (at least better) way so that its content can be more easily and nicely conveyed to the readers.
Our reply: We deeply appreciate your helpful suggestions. We have reconsidered the matter in depth based on your comments. Because Table 6 is intended to reinforce the arguments in the Discussion and contains material that requires interpretation, we felt it was appropriate to place it in the Discussion.
However, some parts were difficult to understand; thus, we have added and revised them to make the overall argument easier to understand.
#Generally, this Ms is well-written with a good command of English. But a few potentially unfavorable points or editing errors, although they may be minor, are noted. Please check the Ms again.
Our reply: We deeply appreciate your positive evaluation of our study. We have re-checked the manuscript and modified typos and editing errors.
Ex:
-line 96 ‘bioproducts’ whereas Table 1 ‘Bio products’: Whether a space is present or not? A term meaning the same thing should be identical.
Our reply: We deeply appreciate your comment. We have modified “bioproduct” and “bio product” to a more adequate word, “biological product.”
-line 103 ‘enrollment’ whereas Table 1 ‘Enrolment’: I know both ‘enrollment’ and ‘Enrolment’ are grammatically correct. But just one form of spelling for a thing is more advisable???
Our reply: We deeply appreciate your comment. We have eliminated terminology discrepancies, modifying “enrollment” to “enrolment”.
-line 122 to 123: (RANO) [11] in 6 and (iRANO) [12] in 1 (?) (*This is judging from Table 3-a)
-Table 3-a, b:
Ex in Table 3-a: *The ORR assessment methods for all seven were RANO in six and iRANO in one. (?)
Our reply: We deeply appreciate your comment. We eliminated terminology discrepancies to the extent possible. Please note that single-digit numbers were not spelled-out when used without units of measurement, as per academic writing standards. In all other cases, they were spelled-out.
-Neurooncology or Neuro-oncology (?) throughout the Ms including the tables.
Our reply: We deeply appreciate your comment. We have eliminated terminology discrepancies, modifying to “Neuro-Oncology”.
-line 193: Pes >>> PEs
-etc., throughout the Ms
Our reply: We deeply appreciate your comment. We have modified “Pes” to PEs. In addition, we have rechecked and corrected inconsistencies in expression throughout the paper.
Reviewer 5 Report
Comments and Suggestions for Authors
The article examines recent trends in the setting of efficacy endpoints in Phase II glioblastoma trials conducted between 2020 and 2022, and compares these with trials from 2017 to 2019.The study investigates how the criteria for evaluating treatment efficacy (specifically primary and secondary endpoints) have changed over time. The results suggest a shift in the primary endpoint, with a notable decline in the use of the objective response rate (ORR) and an increasing variety in the selection of endpoints.
Strengths of the Study:
Relevance: The study focuses on a recent period (2020–2022), offering valuable insights into the rapidly evolving landscape of glioblastoma clinical research, influenced by new therapeutic approaches and innovative endpoint criteria.
Large Number of Studies: The article is based on a broad dataset of 88 Phase II trials, enhancing the representativeness and robustness of the findings, allowing for a comprehensive analysis of trends and shifts in clinical practice.Comprehensive Analysis:The study not only considers primary endpoints (such as PFS and OS) but also includes secondary endpoints and the specific use of time-to-event (TTE) endpoints, leading to a nuanced understanding of efficacy evaluation and its development over time.
The statistical analysis employs appropriate methods (e.g., Wilcoxon test and Fisher's Exact Test) for data analysis, demonstrating a rigorous methodological approach that supports the validity of the conclusions.The article also highlights future challenges and the ongoing evolution of endpoint criteria, particularly regarding the adoption of newer methods like the RANO criteria and the integration of immunotherapies, providing a valuable perspective for the direction of clinical research.
The study's limitations include its inability to be compared with other studies due to its reliance on a single database, Clarivate's Cortellis™ Clinical Trial Intelligence. This database could introduce biases as it excludes relevant studies from other sources and non-English publications.A more comprehensive analysis could be achieved by utilizing a broader dataset.Additionally, the study does not adequately address treatment effects, despite examining endpoint criteria. There is a lack of discussion regarding the actual effectiveness of the treatments themselves. It would be beneficial to see how the chosen endpoints correlate with real treatment outcomes to assess the relevance of endpoint selection.Omission of Protocol Variations: The study does not consider how variations in study protocols (e.g., combination therapies, treatment regimens) may have influenced the choice of endpoints. An analysis of this could provide further insights into the rationale behind selecting certain endpoints.
Overall, this study is commendably well-conducted and well-structured, shedding light on the evolving nature of endpoint criteria in Phase II glioblastoma trials over the past few years.The findings are highly relevant to both clinical researchers and regulatory bodies, as they provide valuable information on the development of efficacy evaluation in this challenging cancer type.
Notwithstanding the aforementioned limitations, the study makes a substantial contribution to the enhancement of methodology in glioblastoma research and contributes to the ongoing discourse on the optimal approaches for assessing treatment efficacy.The study's strengths, particularly the extensive sample size and the meticulous statistical analysis, outweigh the limitations.
Author Response
Reviewer5
The article examines recent trends in the setting of efficacy endpoints in Phase II glioblastoma trials conducted between 2020 and 2022, and compares these with trials from 2017 to 2019.The study investigates how the criteria for evaluating treatment efficacy (specifically primary and secondary endpoints) have changed over time. The results suggest a shift in the primary endpoint, with a notable decline in the use of the objective response rate (ORR) and an increasing variety in the selection of endpoints.
Strengths of the Study:
Relevance: The study focuses on a recent period (2020–2022), offering valuable insights into the rapidly evolving landscape of glioblastoma clinical research, influenced by new therapeutic approaches and innovative endpoint criteria.
Large Number of Studies: The article is based on a broad dataset of 88 Phase II trials, enhancing the representativeness and robustness of the findings, allowing for a comprehensive analysis of trends and shifts in clinical practice.Comprehensive Analysis:The study not only considers primary endpoints (such as PFS and OS) but also includes secondary endpoints and the specific use of time-to-event (TTE) endpoints, leading to a nuanced understanding of efficacy evaluation and its development over time.
The statistical analysis employs appropriate methods (e.g., Wilcoxon test and Fisher's Exact Test) for data analysis, demonstrating a rigorous methodological approach that supports the validity of the conclusions.The article also highlights future challenges and the ongoing evolution of endpoint criteria, particularly regarding the adoption of newer methods like the RANO criteria and the integration of immunotherapies, providing a valuable perspective for the direction of clinical research.
The study's limitations include its inability to be compared with other studies due to its reliance on a single database, Clarivate's Cortellis™ Clinical Trial Intelligence. This database could introduce biases as it excludes relevant studies from other sources and non-English publications.A more comprehensive analysis could be achieved by utilizing a broader dataset.Additionally, the study does not adequately address treatment effects, despite examining endpoint criteria. There is a lack of discussion regarding the actual effectiveness of the treatments themselves. It would be beneficial to see how the chosen endpoints correlate with real treatment outcomes to assess the relevance of endpoint selection.Omission of Protocol Variations: The study does not consider how variations in study protocols (e.g., combination therapies, treatment regimens) may have influenced the choice of endpoints. An analysis of this could provide further insights into the rationale behind selecting certain endpoints.
Overall, this study is commendably well-conducted and well-structured, shedding light on the evolving nature of endpoint criteria in Phase II glioblastoma trials over the past few years.The findings are highly relevant to both clinical researchers and regulatory bodies, as they provide valuable information on the development of efficacy evaluation in this challenging cancer type.
Notwithstanding the aforementioned limitations, the study makes a substantial contribution to the enhancement of methodology in glioblastoma research and contributes to the ongoing discourse on the optimal approaches for assessing treatment efficacy.The study's strengths, particularly the extensive sample size and the meticulous statistical analysis, outweigh the limitations.
Our reply: We deeply appreciate your positive evaluation of our study. We hope that this study will shed light on the evolving nature of efficacy endpoints in Phase II clinical trials for glioblastoma over the past few years and provide evidence that will contribute to the development of efficacy endpoint settings in this challenging disease. As this study does have limitations, as mentioned in the comment, I have added a note to “4.3. Strengths and limitations of the study.”
Round 2
Reviewer 1 Report
Comments and Suggestions for Authors
I would like to thank the Authors for the revision of the paper. It is now, in my opinion, suitable for publication.
Reviewer 4 Report
Comments and Suggestions for Authors
The MS has been faithfully revised. The revised one is much easier to understand.
Comments on the Quality of English LanguageGood.